# Reliability and Validity of the 10-Meter Walk Test (10MWT) in Adolescents and Young Adults with Down Syndrome

**DOI:** 10.3390/children10040655

**Published:** 2023-03-30

**Authors:** Juan Luis Sánchez-González, Inés Llamas-Ramos, Rocío Llamas-Ramos, Francisco Molina-Rueda, María Carratalá-Tejada, Alicia Cuesta-Gómez

**Affiliations:** 1Department of Nursing and Physiotherapy, Faculty of Nursing and Physiotherapy, Universidad de Salamanca, C/ Donantes de Sangre s/n, 37007 Salamanca, Spain; 2University Hospital of Salamanca, Paseo de San Vicente, 182, 37007 Salamanca, Spain; 3Department of Physical Therapy, Occupational Therapy, Rehabilitation and Physical Medicine, Faculty of Health Sciences, Rey Juan Carlos University, 28933 Madrid, Spain

**Keywords:** 10-Meter Walk Test, Down syndrome, gait, validity, reliability, measurement properties

## Abstract

People with Down syndrome (DS) have gait deficits because motor milestones are usually reached later. Decreased gait speed or reduced stride length are some of the main deficits. The main objective of the present work was to assess the reliability of the 10-Meter Walk Test (10MWT) in adolescents and young adults with DS. The objective has been to analyze the construct validity of the 10MWT with the Timed up and go (TUG) test. A total of 33 participants with DS were included. Reliability was verified by intraclass correlation coefficient (ICC). The agreement was analyzed by the Bland-Altman method. Finally, construct validity was evaluated through Pearson correlation coefficient. The 10MWT intra-rater and inter-rater reliability were good (ICC between 0.76 until 0.9) and excellent (ICC > 0.9), respectively. The minimal detectable change for intra-rater reliability was 0.188 m/s. Also, it has demonstrated moderate construct validity (r > 0.5) considering the TUG test. The 10MWT has shown high intra- and inter-rater reliability and validity in adolescent and adults with SD and a moderate construct validity between TUG test and 10MWT.

## 1. Introduction

Down syndrome (DS) is one of the most common genetic disorders, caused by trisomy 21. It has an incidence of 13.65 per 10,000 live births annually in the United States [1,2] and 1 per every 1000–1100 live births in the world [3].

Children with DS present cognitive disability, difficulties in functional skills and social interaction, orthopedic and neuromuscular problems such as hypotonia or ligamentous hyperlaxity, which usually results in an atypical motor development, loss of balance and, in some children, obesity and congenital heart disease [1,2,4,5]. Several authors as Galli et al. [5] focused their study in the relation of foot abnormalities (being the flat food the most common alteration) and its relation with the gait alterations in children with DS. Besides, these alterations will condition the development of more complex task as jumping, running or climbing stairs. In addition, independent walking represents an important achievement in children’s relationships because of the influence it has on social development. Delays in achieving it will have social and cognitive consequences that may hinder interaction with others and with the environment [1].

Babies with DS usually show atypical antigravity control in both legs and the neck, and, in terms of motor development, motor milestones are usually achieved later [1,3]. They reach autonomous sitting at 11 months; these children begin to crawl between 12.2 and 17.3 months, they also maintain standing at 17 months, and they begin autonomous walking at around 24 months (between 18 and 36 months); while typically non-disabled children achieve this independent walking between 11 and 15 months [1,2,4,6]. Children with DS need more time to learn movements as their complexity increases and the requirements for maintaining the posture, the balance, the weight bearing, and producing muscle force increase [2]. Because of ligamentous laxity and hypotonia, children with DS present certain structural alterations such as spinal and/or hip instabilities, scoliosis and foot problems [3]. Due to these structural alterations and the delay in motor development, it is observed that there are associated social and cognitive difficulties, which limit and hinder their participation, their ability to explore and their activities of daily living [1] and contribute to the decrease in bone mass and/or obesity [3]. All these alterations produced by the characteristics of this chromosomal abnormality, together with the possible complications that can occur in this disease and the delay of motor experiences in childhood explains that adolescents and adults with DS have gait disturbances [5,7]. In this sense, those present walking deficits that differs the physiological standard. Typical changes in DS walking comprise spatiotemporal parameters, such as decreased gait speed, longer double support phase, reduced step length, and increased step width [8,9].

For these reasons, it is necessary to have valid and reliable tools to assess the gait of adolescents and young adults with DS. Some tests such as the TUG test have been used in the DS population and have shown that they can be used as a screening test for functional mobility of children and adolescents with DS [10]. The 10-Meter Walk Test (10MWT) has been used to assess walking ability in pediatric patients in clinical practice. This test has the potential to provide valuable clinical information on gait abilities. It is safe, easy, and inexpensive to administer [11]. The 10MWT has demonstrated excellent reliability in many conditions including healthy adults and children and young adults with neuromuscular disease [12,13].

The hypothesis of the study has been that the 10MWT will be a valid and reliable tool to assess gait parameters in adolescents and young adults with DS. The main objective of the present work has been to assess the validity and reliability of the 10MWT in adolescents and young adults with DS. The objective has also been to analyze the construct validity of the 10MWT with the TUG test, since this test had been previously validated in the DS population [10].

## 2. Materials and Methods

### 2.1. Sample Size

The sample size was calculated based on Walter et al. [14]. It was estimated size using Intraclass Correlation Coefficient (ICC) and the number of raters. Considering a minimally acceptable ICC (*p*0) of 0.4, and an expected ICC (*p*1) of 0.7, and following the contingency tables of Walter et al. [14], which results in a minimum sample of 33 subjects for intrarater reliability and 21 subjects for interrater reliability.

### 2.2. Participants

Subjects with DS were recruited from the different Spanish associations focused on DS children. The inclusion criteria were: having a diagnosis of DS with an age between 8 and 30 years old, having the ability to understand verbal instructions (a score ≥20 in the Mini-Mental Test for children), having the ability to walk a minimum of 500 m independently (with or without walking aids or orthoses) and having the ability to sit and stand independently (with or without walking aids or orthoses). Subjects were excluded if they presented a diagnosis of any cardiovascular, musculoskeletal, respiratory and/or metabolic disease, or other conditions that may interfere with the implementation of the study. The local ethics committee of the University of Salamanca approved the protocol (2 June 2022; n° 800) and Declaration of Helsinki guidelines have been followed.

### 2.3. Procedure

Children with a DS diagnosed were evaluated in two different days with an interval between the assessments between 5 and 7 days, following the studies of Germanotta et al. [15,16]. Prior to the evaluations, one of the researchers verified that the subjects met the inclusion criteria to be included. In addition, they were given the Mini-Mental Test to verify that they had the minimum score necessary to participate in the study.

Moreover, the information sheet and the informed consent were given to the parents or legal representative of those minor subjects, prior to the evaluations. In addition, an informed consent adapted to those students over 14 years of age was given. All the tests were carried out with comfortable clothes that allowed movement, and parents or legal guardians could be present. All sanitary measures were taken to guarantee hygiene and safety in the context of COVID-19.

The following evaluation tools were used:

10MWT: To perform this test, the subject walked without aids at comfortable gait speed a distance of 10 m. Time was measured, with the stopwatch tool of the smartphone, while the individuals walked the set distance (6 m). Three trials were administered at the patient’s comfortable walking speed. The 3 trials were averaged and the gait speed was documented in meters/second [11]. Patients were allowed to take the trial test once, to check that they had understood what to do.

TUG test: it is a simple test that consists of the subject getting up from a chair with an armrest from the sitting to the bipedal position, walking three meters, turning, returning, and sitting in the chair again. The variable measured is the total time (in seconds) taken to complete the test. Time was measured, with the stopwatch tool of the smartphone. The test was performed according to the modifications for children described in the study by Williams et al. [17]. The height of the chair was modified in each patient so that the feet were in contact with the floor with a knee flexion of 90°. For each TUG test evaluation, three measures are recommended, and the result is the shortest time obtained. In addition, the TUG test can be used to evaluate balance and functional mobility in children and adolescents with Down syndrome [10,17].

For inter-rater reliability, two different evaluators carried out a first evaluation of the 10MWT at the same time. For intra-rater reliability, the 10MWT was repeated identically on the second day of assessment, by the same evaluator on the first day. For construct validity analysis, the TUG test was used, which was administered on the second day, after performing the 10MWT and leaving a rest period between both tests. The evaluators who carried out the tests were qualified and trained in carrying out both tests, both the 10MWT and the TUG test. The instructions given to the participants were always carried out by the same evaluator to avoid bias.

### 2.4. Statistical Analysis

The statistical analyses were conducted SPSS software for Windows (SPSS Inc., Chicago, IL, USA; Version 26.0). It was checked whether the variables followed a normal distribution through the Shapiro-Wilk test.

The intraclass correlation coefficient (ICC) was used to analyze the intrarater reliability of the test, its 95% confidence interval, using a mixed effects model and absolute agreement [18,19]. The ICC value ranges from 0 to 1, being classified as poor (<0.4); moderate (0.40–0.70); good (0.70–0.90); or excellent reliability (>0.90) [20].

Additionally, based on data from the two observation days, the minimal detectable change of the 10MWT was calculated using the following formula: MDC95 = 1.96 *√2 *SEM. The SEM is the standard error of measurement (SEM = SD_diff_√1-ICC), where SD_diff_ is the standard deviation of the differences from test sessions 1 and 2 and ICC is the reliability obtained for the intra-rater reliability [21].

A Bland-Altman analysis was performed with 95% limits of agreement to assess intra- and inter-rater reliability. The bias and the limits of agreement are shown in the graphs of the recorded parameter. The mean score is represented on the x-axis, and the difference between evaluators and sessions (mean of the differences) is represented on the y-axis (mean of the difference ± 1.96 standard deviation). The width of the limits of agreement and the distance of the mean of the differences from zero can be used to interpret the errors between measures [19,22].

The construct validity between 10MWT and TUG test was assessed through the Pearson correlation coefficient. We use the following categories and terminology from Salter et al. [23]: r ≥ 0.75, excellent correlation; r = 0.40–0.74, moderate correlation; and r ≤ 0.40, poor correlation. The significance level was set to 0.05 for all tests.

## 3. Results

A total of 33 subjects participated in the study. The sample was composed of 18 men and 15 women, with a mean age of 18.59 (±6.21) years old, mean weight of 52.56 (±14.25) kg, mean height of 148.87 (±13.88) cm and average body mass index of 23.29 (±4.50).

The 10MWT intra-rater and inter-rater reliability were good (ICC between 0.76 until 0.9) and excellent (ICC > 0.9), respectively (Table 1 and Table 2). The SEM for intra-rater reliability was 0.068 m/s and the MDC was 0.188 m/s.

According to the Bland-Altman plots (Figure 1), the mean of the differences is 0.027 m/s with the limits of agreement (LOA) varying between −0.33 to 0.38 for intra-rater reliability. For inter-rater reliability, the mean of the differences was 0.0047 m/s with LOA ranging from 0.08 to 0.09.

Finally, the construct validity between TUG test and 10MWT was moderate (r = 0.55) (Table 3).

## 4. Discussion

The present study contributes to the literature by providing evidence that the 10MWT exhibits reliability and construct validity for the measurement of gait velocity in children with DS. The aim of the present study has been to evaluate the intra- and inter-rater reliability of the 10MWT and the construct validity between the 10MWT and the TUG test in adolescent and young adults with SD. The use of the 10MWT could provide objective and quantitative data to assess the atypical gait pattern described in subjects with DS.

Children with DS have hypotonia, joint hypermobility and decreased strength that cause alterations in in balance, coordination, and walking [24]. These alterations cause gait deficits at older ages. Several authors suggested that the delayed motor development observed in DS subjects affect the postural control and the walking speed, generating instability during and atypical patterns during gait [7,24]. In this sense, Zago et al., in a systematic review, showed that subjects with DS have spatiotemporal abnormalities in gait: lower speed and stride length and higher stride width. In addition, they walk with increased hip and knee flexion and reduced plantar flexion during the stance period of the gait cycle. Specifically, they exposed that the reduced speed during the gait pattern and the quality of postural reactions lead to compensatory movements in the phases of the gait cycle, which in turn will generate an alteration in the acquisition of adequate patterns to achieve a functional and stable gait. These patterns will generate negative effects on the correct performance of activities of daily living [7].

Foot alterations as fat food deformity could influence the gait pattern. For that reason, Galli et al. [5] investigated this deformity in children with DS versus children without this condition using a 3D gait analysis with an optoelectronic system with force platforms and video recording. They showed that this population ankle plantarflexion movement and the ankle power during the stance were different between groups, being lower in children with flat foot. This results support the conclusion that children with DS has a walking less efficient and less functional that people without it.

The use of orthoses to prevent secondary complications are well documented in the literature [1,5], even achieving immediate positive results [25]. Martin [26] tested a supramleolar orthosis to reduce the pronation caused by the hypotonia that these children with DS have, during 10 weeks for standing, jumping, walking and running conditions. Her results were positive with immediate and long-term effects improving the postural stability using this flexible orthosis. Several studies have also tried to improve gait patterns with foot and ankle orthosis or partial body weight supported treadmill training with positive results [3,25,26]. Specifically, Looper et al. showed an 8 min training, 5 days a week, makes easier the independent walking onset in DS children [1]. Unfortunately, as this was not the objective of the study, we did not take into account the use or not of orthosis, or the type of orthosis but the ability to perform an independent march with or without them.

Our results showed that the 10MWT has good intra-rater and excellent inter-rater reliability for measuring walking speed in adolescent and adults with DS. These results are consistent with previous studies. The 10MWT has demonstrated an excellent intra-rater reliability in adults with traumatic brain injury and excellent inter-rater reliability in healthy subjects; Watson et al. found a CI for agreement of 95% ranged from −0.38 to + 0.38 s for normal young subjects and −0.36 to + 0.49 s for traumatic brain-injured subjects [27] and two years later van Loo et al. performed a study to evaluate the test-retest of this test with excellent results (ICC = 0.95–0.96) and high reliability for step length and width measurement (ICC = 0.91–0.98) [28]. Other authors have shown good validity with the conclusion that it is a quick, simple and inexpensive test to administrate in children with poorer walking ability and cerebral palsy adolescent [12,29] and good intra- and inter-rater reliability in children with neurological impairments with an ICC of 0.81 (95%) ranged from 0.65–0.90 [30,31].

The Bland-Altman plots showed a disagreement varying between −0.33 to 0.38 m/s for intra-rater reliability and between 0.08 to 0.09 m/s for inter-rater reliability. The MDC for intra-rater reliability was 0.188 m/s. The MDC represent the limit for the smallest change that indicates a real improvement in reaching performance in a single individual [32] (in the present study, an individual who had DS). Estimates of MDC have not been published for 10MWT in people with DS, but our data are consistent with the MDC reported in other populations. At comfortable speed, the MDC for 10MWT was 0.18 m/s in subjects with Parkinson’s Disease (Hoehn and Yahr 1–4, median score 2) [33], 0.13 m/s in Spinal Cord Injury (incomplete < 12 months) [34], 0.11 m/s in acute stroke [35], 0.18 m/s in chronic stroke [36] and 0.26 m/s in multiple sclerosis [37].

To our knowledge, this is the first study that has used the 10MWT to assess walking velocity in young and adolescent with DS while studying construct validity with the TUG test. The TUG test is developed to evaluate functional alterations in elderly, being a good predictor of the ability of the patients to go outside alone safely [38]; however, this tool has been shown to be reliable in young children up to 3 years of age, demonstrating that the child understands the instructions and the test can be performed perfectly without behavioral variation [17]. Different authors have demonstrated a good or excellent reliability for the TUG test in pediatric population such as children aged between 3 and 9 years old without disability [17], children aged between 8 and 19 years old with cerebral palsy and spina bifida [17] and children with DS [7]. Our specific results for TUG test showed that the correlation between the 10MWT and the TUG test was moderate (r = 0.55). The construct validity of the 10MWT demonstrated that this tool has a convergent correlation with the balance and functional mobility measured with the TUG test in adolescent and young individual with DS. Given the evident correlation between these constructs, we can note the importance of employing the 10MWT in a clinical situation because gait velocity could have a predictive value for balance and functional mobility in subjects with DS. In addition, this gait assessment can be used to guide the treatment lines for this population.

These results bring us future investigation lines due to the low cost, easy and quick application that this study demonstrated. Another population and larger sample sizes will be implemented. The 10MWT would allow researchers to evaluate and measure the evolution of children with neurological disorders with a valid and reliable scale in Spanish.

### Limitations

The present study has several limitations that must be considered. In the first place, there was no gender distinction, an important aspect since according to Galli et al. [5], men and women present different patterns, especially in terms of kinematic parameters, as also confirmed by the studies by Pau et al. [39], Zago et al. [40] and Cimolín et al. [41].

Another aspect to highlight is the possible existence of factors that can influence gait such as mood and motivation [42]. Considering the difficulties presented by these patients, some authors have paired the subjects with another person to perform the tests [43,44], an aspect that was not implemented in the present study, where the subjects performed the tests independently.

Finally, in the present study the 10MWT test has only been implement in children with DS, other neurological diseases or pathologies with alterations in the gait cycle could benefit from the use of this tool.

## 5. Conclusions

The 10MWT has good intra-rater and excellent inter-rater reliability for assessing gait velocity in adolescent and young individual with DS. The minimal detectable change for intra-rater reliability was 0.188 m/s. In addition, the 10MWT demonstrated a moderate construct validity with the TUG test. The 10MWT test has showed to be valid and reliable in Spanish language and children with Down Syndrome.

## Figures and Tables

**Figure 1 children-10-00655-f001:**
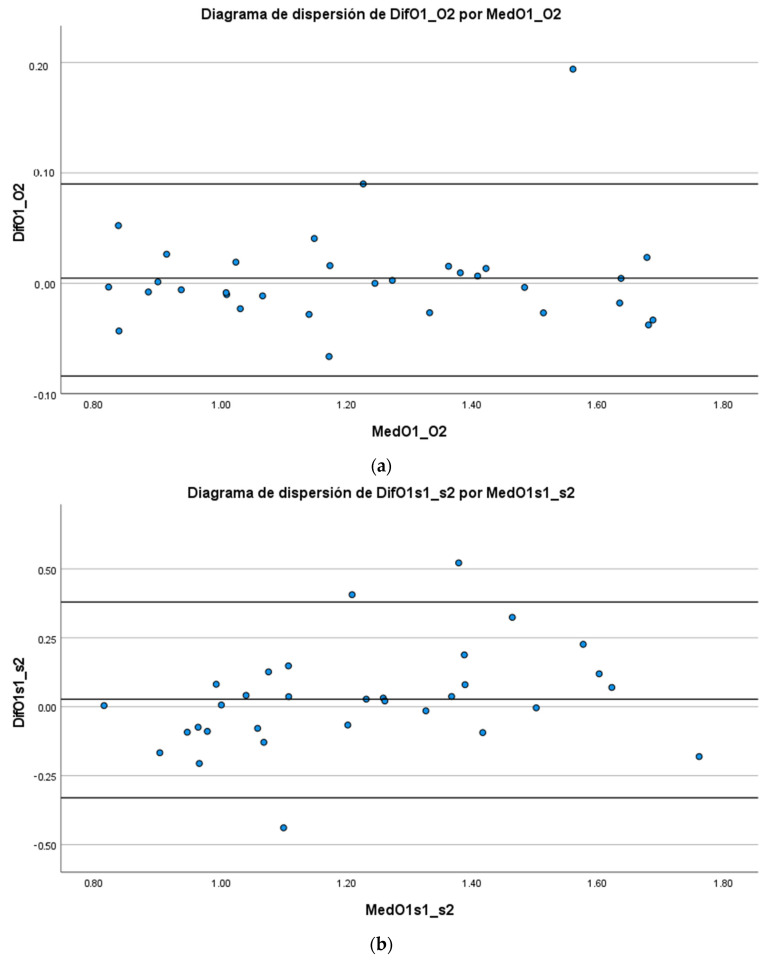
Bland-Altman plots. (**a**) Intra-rater reliability; (**b**) Inter-rater reliability; The bias and limits of agreement (black lines) are displayed for 10MWT (seconds) measurements. The mean score is represented on the x-axis and the difference between sessions (mean of the differences) is represented on the y-axis (mean difference ± 1.96 SD).

**Table 1 children-10-00655-t001:** Intra-rater reliability of the 10MWT.

	Session 1	Session 2	ICC	95% CI	*p*-Value
10MWT	1.23 (0.28)	1.20 (0.22)	0.857	0.987 to 0.997	<0.01 *

Data are expressed in mean and standard deviation. ICC: Intraclass correlation coefficient, CI: Confidence Interval. * *p*-value < 0.05.

**Table 2 children-10-00655-t002:** Inter-rater reliability of the 10MWT.

10MWT		RRater 1 vs. Rater 2		
Rater 1	Rater 2	ICC	95% CI	*p*-Value
1.23 (0.28)	1.22 (0.27)	0.994	0.989 to 0.997	<0.01 *

Data are expressed in mean and standard deviation. ICC: Intraclass correlation coefficient, CI: Confidence Interval. * *p*-value < 0.05.

**Table 3 children-10-00655-t003:** Validity of the 10MWT.

			Rater 1 vs. TUG			Rater 2 vs. TUG		
Rater 1	Rater 2	TUG	r	*p*-Value	95% CI	r	*p*-Value	95% CI
5.12 (1.19)	5.12 (1.18)	7.63 (1.35)	0.553	<0.01 *	−0.756 to −0.253	0.522	0.002 *	−0.737 to −0.212

Data are expressed in mean and standard deviation. R: Pearson correlation. CI: Confidence Interval. * *p*-value < 0.05.

## Data Availability

The data that support the findings of this study are available on reasonable request from the corresponding author.

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
