# Peer review of "Reliability and Validity of the 10-Meter Walk Test (10MWT) in Adolescents and Young Adults with Down Syndrome"

_children, 2023, doi:10.3390/children10040655_

Round 1

Reviewer 1 Report

Dear Authors

You have written an interesting paper focused on the assessment of the 10-Meter Walk Test (10MWT) reliability in 16 adolescents and young adults with Down Syndrome.

However, some parts need to be addressed for greater clarity. 

Introduction - 

Line 39 - childhood - be more specific as there are multiple stages of children's development. Be specific. Also this sentence needs to be backed up by a reference. Amend

Describe which tests have been used in DS population or similar populations and then present the importance of 10MWT. Also, back this up with more references - especially sentences in lines 45-47. 

You didn't present a clear rationale why will you compare 10MWT to TUG test / is TUG a gold standard? 

Amend the introduction accordingly.

Methods

Well done for reporting the calculation of sample size.

When was the Mini-Mental Test done? How did you check for the inclusion criteria? report

Report basic data of participants' gender, age, height, weight, physical activity level, etc.

Were there any practice trials before the testing of both tests? Add info

Tug - report the height of the chair

How and with what device was time measured?

On how many participants was the inter and intra-rater reliability done? Also, you mention 

Why wasn't the TUG and 10MWT test order chosen randomly? This could have an effect on the results.

Correct font size and spelling in Tables 1 and 2

The discussion could be better formed and include some comparison of results with other studies.

Overall the study needs some more work. Therefore, I recommend a major revision.

Kind regards

Author Response

Reviewer 1

Dear Authors

You have written an interesting paper focused on the assessment of the 10-Meter Walk Test (10MWT) reliability in 16 adolescents and young adults with Down Syndrome.

However, some parts need to be addressed for greater clarity.

Thank you very much for your positive comments.

Introduction -

Line 39 - childhood - be more specific as there are multiple stages of children's development. Be specific. Also this sentence needs to be backed up by a reference. Amend

Thank you very much for your comment. We have added the references.

Describe which tests have been used in DS population or similar populations and then present the importance of 10MWT. Also, back this up with more references - especially sentences in lines 45-47.

We have added the information required

You didn't present a clear rationale why will you compare 10MWT to TUG test / is TUG a gold standard?

Thank you very much for your question. We have compared the 10MWT with the TUG, because the TUG has already been previously validated in the SD population, as well as being a fast and reliable simple tool. We have added the reference of its validation and a sentence of this

Amend the introduction accordingly.

Methods

Well done for reporting the calculation of sample size.

Thank you very much for your comment

When was the Mini-Mental Test done? How did you check for the inclusion criteria? Report

Thank you very much for your comment. We have added this information

Report basic data of participants' gender, age, height, weight, physical activity level, etc.

Thank you very much for your comment. This data is reported in results section

Were there any practice trials before the testing of both tests? Add info

Thank you very much for your comment. We have added this information

Tug - report the height of the chair

Thank you very much for your comment. We have reported this data

How and with what device was time measured?

Thank you very much for your comment. We have explained it

On how many participants was the inter and intra-rater reliability done? Also, you mention

Thank you very much for your comment. In all the participants, 33 in total the inter and intra-rater reliability were done. It is mention in results section

Why wasn't the TUG and 10MWT test order chosen randomly? This could have an effect on the results.

Thank you very much for your comment. It was decided to perform the 10MWT first and then the TUG, leaving a rest interval between both tests. We have added this information

Correct font size and spelling in Tables 1 and 2

Thank you very much for your comment. We have corrected the three tables

The discussion could be better formed and include some comparison of results with other studies.

Thank you very much for your comment. A results comparison has been added.

Overall the study needs some more work. Therefore, I recommend a major revision.

Kind regards

Reviewer 2 Report

The article ‘Reliability and validity of the 10-Meter Walk Test (10MWT) in adolescents and young adults with Down syndrome’ investigates the reliability and validity of the 10MWT in a specific population of individuals with Down syndrome. The authors provide a clear background and rationale for the study, which is well-supported by the literature.

However, there are some potential limitations to consider. Firstly, the sample size was calculated based on another study, which may not have taken into account certain factors such as age, sex, and severity of Down syndrome. These factors could also have an impact on the results, which means that a more diverse sample size might be required for greater accuracy.

Additionally, the study did not consider fatigue or learning effects, which could affect the reliability of the 10MWT. As such, the authors may wish to include additional measures or considerations in future research to better account for these factors.

Another potential area for improvement is in the methodology section. Specifically, the authors may wish to provide more information about the raters' experience and training, as well as any standardisation of instructions given to participants. This could help to ensure greater consistency and reliability in the results obtained.

Overall, the article presents valuable findings for clinicians and researchers working with individuals with Down syndrome, but it is important to consider these potential limitations and call for further research to confirm the validity and reliability of the 10MWT in larger and more diverse populations.

Author Response

Reviewer 2

The article ‘Reliability and validity of the 10-Meter Walk Test (10MWT) in adolescents and young adults with Down syndrome’ investigates the reliability and validity of the 10MWT in a specific population of individuals with Down syndrome. The authors provide a clear background and rationale for the study, which is well-supported by the literature.

We would like to thank the reviewer for taking the time to thoroughly review our manuscript and provide recommendations for improving it.

However, there are some potential limitations to consider. Firstly, the sample size was calculated based on another study, which may not have taken into account certain factors such as age, sex, and severity of Down syndrome. These factors could also have an impact on the results, which means that a more diverse sample size might be required for greater accuracy.

Thank you for your comment, sample size was calculated based on Walter, S.D.; Eliasziw, M.; Donner, A. Sample size and optimal designs for reliability studies. Stat Med 1998; 17:101-110. That consider the factors necessary to calculate the sample size in studies of reliability studies. That is why we believe that the total number of subjects included in the study is optimal.

Additionally, the study did not consider fatigue or learning effects, which could affect the reliability of the 10MWT. As such, the authors may wish to include additional measures or considerations in future research to better account for these factors.

Thanks for the suggestion, regarding fatigue this factor was considered and for this reason the patients were allowed to rest between 10MWT and the TUG, so that it would not affect the test times, we have explained it in the material and methods section. Regarding the learning factor, only the 10MWT test that is carried out in two days can be affected and since it is simply walking at your normal walking speed, we do not consider that there may be a learning factor.

If it is considered that the learning factor may lie in repeating each test three attempts, as described in the rules for carrying them out, to avoid this, the averages of the three tests are made.

Another potential area for improvement is in the methodology section. Specifically, the authors may wish to provide more information about the raters' experience and training, as well as any standardisation of instructions given to participants. This could help to ensure greater consistency and reliability in the results obtained.

Thank you for the suggestion, we have clarified this in material and methods section.

Overall, the article presents valuable findings for clinicians and researchers working with individuals with Down syndrome, but it is important to consider these potential limitations and call for further research to confirm the validity and reliability of the 10MWT in larger and more diverse populations.

Thank you very much for the suggestion, we hope that the changes made provide more clarity and improvements to the manuscript for its acceptance.

Reviewer 3 Report

The authors are to be congratulated for the originality and challenge faced to develop this research. Small adjustments deserve to be observed and made. I bring my contributions to evaluate carefully, but I reinforce that nothing takes away the merit of what has already been developed so far.

Introduction

Line 43 - I suggest pointing out the relationship of gait with other important aspects such as cognitive ability, respiratory condition and psychomotor functions.

line 43 - I suggest briefly highlighting which instruments for functional capacity analysis have been used in this population and their challenges and limitations in their execution.

What is the hypothesis formulated by the authors?

Sample size 

Perhaps the sample calculation could also be based on a study that evaluated some functional aspect specifically for this population. I suggest evaluating this possibility. 

Participants

Line 64- Subjects with unilateral DS ????  It's unclear. 

I suggest including information about the Consent Form and making some reference about the Declaration of Helsinki.

I think the information on ethical issues is better presented in Participant than in Procedures.

Discussion 

I suggest that the condition (ability to understand verbal instructions: a score ≥ 20 in the Mini-Mental Test for children) needs to be better addressed in the discussion. Are there previous studies determining this parameter? Does this compromise the execution of the test when minor children and adolescents are compared?  Does the level of impairment of mental capacity prevent the execution of the test?

It is strongly indicated that the authors point out the strength of this work and the future perspectives, considering that this is another tool that we will have available to evaluate the development of these children and adolescents with DS.

Author Response

Reviewer 3

The authors are to be congratulated for the originality and challenge faced to develop this research. Small adjustments deserve to be observed and made. I bring my contributions to evaluate carefully, but I reinforce that nothing takes away the merit of what has already been developed so far.

Thank you for your comment.

Introduction

Line 43 - I suggest pointing out the relationship of gait with other important aspects such as cognitive ability, respiratory condition and psychomotor functions.

We have added this information.

line 43 - I suggest briefly highlighting which instruments for functional capacity analysis have been used in this population and their challenges and limitations in their execution.

Thank you for your comment, we have described in line 54 other test that have been validated in this population.

What is the hypothesis formulated by the authors?

Thank you for your suggestion. Hypothesis has been added.

Sample size 

Perhaps the sample calculation could also be based on a study that evaluated some functional aspect specifically for this population. I suggest evaluating this possibility. 

Thank you for your suggestion. Sample size was calculated based on Walter, S.D.; Eliasziw, M.; Donner, A. Sample size and optimal designs for reliability studies. Stat Med 1998; 17:101-110. That consider the factors necessary to calculate the sample size in studies of reliability studies. That is why we believe that the total number of subjects included in the study is optimal.

Participants

Line 64- Subjects with unilateral DS ????  It's unclear. 

Thank you for this comment. This word has been removed to clarify.

I suggest including information about the Consent Form and making some reference about the Declaration of Helsinki.

Thank you for your comment, consent form information is located in lines 83-84 and Helsinki guidelines mention has been added in line 75.

I think the information on ethical issues is better presented in Participant than in Procedures.

Thank you for your suggestion. This data has been moved to participant’s section.

Discussion 

I suggest that the condition (ability to understand verbal instructions: a score ≥ 20 in the Mini-Mental Test for children) needs to be better addressed in the discussion. Are there previous studies determining this parameter? Does this compromise the execution of the test when minor children and adolescents are compared?  Does the level of impairment of mental capacity prevent the execution of the test?

Thank you for your suggestion. We do not believe that this condition has compromised the execution of the study because the ability to understand the orders to carry it out is needed. In fact, we believe that if we had admitted patients with lower scores on this scale, the study results would have been compromised because the patients did not understand the test orders. In this sense, Nicolini-Panisson et al. (Nicolini-Panisson, R.D.; Donadio, M.V. Normative values for the Timed 'Up and Go' test in children and adolescents and validation for individuals with Down syndrome. Dev Med Child Neurol 2014; 56, 490-7) also limits the sample depending on your intellectual capacity

It is strongly indicated that the authors point out the strength of this work and the future perspectives, considering that this is another tool that we will have available to evaluate the development of these children and adolescents with DS.

Thank you for your suggestion. This information has been added.

Round 2

Reviewer 1 Report

Dear Authors,

Thank you for addressing the majority of my questions and suggestions.

However, there are still some parts that need to be addressed:

-In the introduction please check the DS abbreviation as in some cases you use SD.

Report basic data of participants' gender, age, height, weight, physical activity level, etc. / Thank you very much for your comment. This data is reported in results section

This needs to be in the methods and NOT in the results! Move  / Additionally report how did you measure body height and weight.

Why wasn't the TUG and 10MWT test order chosen randomly? This could have an effect on the results. / This needs to be added in the limitations of the study

Kind regards

Author Response

Dear Authors,

Thank you for addressing the majority of my questions and suggestions.

However, there are still some parts that need to be addressed:

-In the introduction please check the DS abbreviation as in some cases you use SD.

Thank you for your suggestion. These words has been corrected.

Report basic data of participants' gender, age, height, weight, physical activity level, etc. / Thank you very much for your comment. This data is reported in results section

This needs to be in the methods and NOT in the results! Move  / Additionally report how did you measure body height and weight.

Thank you for your recommendation, we have added how did we measure body height and weight. As for moving these data to the material and methods section, we do not believe that this has to be changed because in the material and methods section it is exposed how the study was carried out, the variables, the sample, inclusion criteria, exclusion, the sample size, statistical analysis... in short, the methods. And in the results section the sample that has formed part of this study is exposed, therefore we do not consider this change necessary, although we appreciate your comment.

Why wasn't the TUG and 10MWT test order chosen randomly? This could have an effect on the results. / This needs to be added in the limitations of the study

Thanks for the recommendation, but we don't agree that this is a limitation. For a validation study of a measuring instrument, both measurements must be performed under the same conditions to be reproducible and reliable. If we carry out the tests in a randomized manner, we would no longer be doing the validation under the same conditions. In addition, if the instructions of the tests are followed, as we have done, the attempts of each test must be carried out in a row, without breaks, if we randomized each TUG test attempt and 10 MWT, we would no longer be performing the test as their authors order. For this reason, we believe that the study was carried out in the most optimal conditions plus the validation and reliability of the TUG test.

 Thank you very much for your considerations.